# A Genome-Wide Association Study of Nigerien and Senegalese Sorghum Germplasm of *Exserohilum turcicum*, the Causal Agent of Leaf Blight

**DOI:** 10.3390/plants12234010

**Published:** 2023-11-29

**Authors:** Louis K. Prom, Jacob R. Botkin, Ezekiel J. S. Ahn, Mame Penda Sarr, Cyril Diatta, Coumba Fall, Clint W. Magill

**Affiliations:** 1USDA-ARS, Southern Plains Agricultural Research Center, 2765 F & B Road, College Station, TX 77845, USA; 2Department of Plant Pathology, University of Minnesota, St. Paul, MN 55108, USA; botki009@umn.edu; 3USDA-ARS Plant Science Research Unit, St. Paul, MN 55108, USA; 4Centre National de Recherches Agronomiques de Bambey, BP 53 Bambey, Senegal; sarrapenda@gmail.com (M.P.S.); batacyril@yahoo.fr (C.D.); 5Department of Plant Pathology and Microbiology, Texas A&M University, College Station, TX 77843, USA; coumba@tamu.edu (C.F.); c-magill@tamu.edu (C.W.M.)

**Keywords:** sorghum, *Exserohilum turcicum*, leaf blight, incidence, severity, GWAS

## Abstract

In Senegal, sorghum ranks third after millet and maize among dryland cereal production and plays a critical role in the daily lives of millions of inhabitants. Yet, the crop’s productivity and profitability are hampered by biotic stresses, including *Exserohilum turcicum*, causing leaf blight. A total of 101 sorghum accessions collected from Niger and Senegal, SC748-5 and BTx623, were evaluated in three different environments (Kaymor, Kolda, and Ndiaganiao) in Senegal for their reactions against the leaf blight pathogen. The results showed that 11 out of the 101 accessions evaluated exhibited 100% incidence, and the overall mean incidence was 88.4%. Accession N15 had the lowest incidence of 50%. The overall mean severity was 31.6%, while accessions N15, N43, N38, N46, N30, N28, and N23 from Niger recorded the lowest severity levels, ranging from 15.5% to 25.5%. Accession N15 exhibited both low leaf blight incidence and severity, indicating that it may possess genes for resistance to *E*. *turcicum*. Also, the accessions evaluated in this study were sequenced. A GWAS identified six novel single-nucleotide polymorphisms (SNPs) associated with an average leaf blight incidence rate. The candidate genes were found in chromosomes 2, 3, 5, 8, and 9. Except for SNP locus S05_48064154, all five SNPs associated with the leaf blight incidence rate were associated with the plant defense and stress responses. In conclusion, the candidate genes identified could offer additional options for enhancing plant resistance against *E. turcicum* through plant breeding or gene editing.

## 1. Introduction

Sorghum (*Sorghum bicolor* (L.) Moench) is one of the most indispensable and versatile crops in terms of its uses and adaptability, especially in the arid and semi-arid regions of the world, where it plays a critical role in subsistence farming and the daily calorie needs of hundreds of millions of people [1,2,3,4]. In Senegal, sorghum is a vital commodity, food, and feed source for millions of inhabitants, and ranks third behind millet and maize in dryland cereal production [4,5]. In the last 5 years (2017/2018 to 2021/2022), a mean area of planted sorghum of 250,000 ha, as well as 302,000 tons of production, was recorded in Senegal [6]. During the same period, the mean yield in the country was 1.2 tons/ha [6], despite the fact that most of the sorghum crops planted were landraces with low yields; however, some of the advanced/improved varieties developed and released by Institut Sénégalais de Recherches Agricoles/Centre National de Recherches Agronomiques can produce yields of up to 5 tons/ha in experimental stations with high farm inputs [4,7]. Nevertheless, sorghum production and profitability are constrained by biotic stresses in Senegal, including leaf blight, incited by *Exserohilum turcicum* (Pass,) K. J. Leonard & E. G. Suggs [syn. *Helminthosporium turcicum* (Pass.)] [8]. The pathogen also infects teosinte, Sudan grass, gamagrass, and johnsongrass, and the infection is most severe under heavy dew and temperatures ranging from 18 to 27 °C [8]. Studies have shown that *E*. *turcicum* is host-specific, meaning that the maize isolates of the pathogen are non-pathogenic to sorghum [8,9,10]. The reverse is true, resulting in two host-specific strains: *E*. *turcicum* f. sp. *zeae* attacks maize and *E*. *turcicum* f. sp. *sorghi* incites leaf blight on sorghum [9,10]. The host specificity is reported to be conferred by a single gene in the pathogen, *SorA*^+^ in sorghum and *ZeaA^+^* in maize [10]. The leaf blight pathogen can survive on sorghum as mycelia, conidia in plant debris, weed hosts, or chlamydospores without host tissue in the soil [11]. The pathogen can infect both young and old plants, with symptoms characterized by large, elongated, spindle-shaped spots with a straw/grey color, surrounded by pigmented margins, and in older plants, the lesions may have yellowish-to-gray centers with reddish margins [8,11]. Sorghum leaf blight is widely distributed in areas where sorghum is planted, and under severe foliar infection, losses of up to 70% can occur [8,12,13,14]. Prom et al. [15] surveyed 206 sorghum farmers’ production fields in Senegal during the 2019 growing season and noted that 198 fields had leaf blight-infected plants. Resistant sources of the sorghum leaf blight pathogen have been reported. Hepperly and Sotomayor-Ríos [16] evaluated sorghum germplasm in Isabel, Puerto Rico, and identified several sorghum lines, including IS12526, IS12576, IS12606, and IS12638, that possess high levels of resistance. Four lines, Gambella-1107, Seredo, 76Tl#23, and Meko-1, were found to be resistant to *E*. *turcicum* f. sp. *sorghi* when evaluated in two locations in Southern Ethiopia [13].

However, the recent advances made using genomic resources (e.g., molecular markers, genomic mapping software) have made it possible to identify single-nucleotide polymorphisms (SNPs) to locate genes associated with economically important traits such as disease resistance [17]. For instance, Tomar et al. [18] identified eight QTLs associated with spot blotch disease resistance in the wheat genome using 14,063 polymorphic genotyping-by-sequencing markers. A total of 17 QTLs were identified at different genomic locations when wheat cultivars were evaluated in a greenhouse against four pathotypes of the stripe rust pathogen [19], while Liu et al. [20] documented 14 loci associated with stripe rust in multiple locations, as well as 37 loci with a significant association with the disease at all plant developmental stages. Further, Kawicha et al. [21] identified six unique significant SNPs for tomato fusarium wilt resistance located on chromosomes 2, 4, and 7 of the tomato genome. We hypothesized that SNPs associated with the leaf blight response could be identified, and that these SNPs might have a role in host defense. Thus, genome-wide association studies (GWASs) of the incidence and severity of the leaf blight pathogen against Nigerien and Senegalese sorghum germplasms, planted in three environments in Senegal, were conducted.

## 2. Results

The leaf blight incidence rate across the three locations in the Nigerien and Senegalese accessions was high, with the average rate = 88.4 ± 0.76 (Table 1). The lowest incidence rates were observed in N15, N48, and S17. Likewise, Table 2 depicts the leaf blight severity levels from the highest to the lowest accessions. The accessions with the lowest severity levels are from Niger, including N15, N43, N38, N46, N30, N28, and N23. The highest leaf blight severity (75.5%) was recorded in Kaymor. In contrast, many susceptible accessions were from Senegal.

The mean leaf blight incidence rate and severity were calculated based on three different locations: Kaymor, Kolda, and Ndiaganiao. The mean incidence rate in Kolda (mean ± S.E.= 75 ± 1.5) was statistically lower than the other two locations (Kaymor = 96.2 ± 12.8 and Ndiaganiao = 96 ± 13.6 for mean ± S.E.) (Figure 1). Ndiaganiao showed a significantly high severity level (mean ± S.E. = 39.7 ± 0.6) compared to Kaymor ( Mean ± S.E. = 27.9 ± 0.8) and Kolda (mean ± S.E. = 26.3 ± 0.6).

Pearson’s correlation indicates a clear positive correlation between the leaf blight incidence rate and severity (Figure 2). As described in Figure 1, many accessions in Ndiaganiao showed the highest incidence rate and severity level.

A PCA plot (Figure 3) and a dendrogram (Figure 4) generated based on SNP data agreed that four major groups exist among the Nigerien and Senegalese accessions. Figure 4 shows multiple accessions from the two countries that merge and form a group (blue).

Figure 5 and Table 3 show the top SNPs passing the Bonferroni threshold, identified from the GWAS, and their associated genes for the leaf blight incidence rate. Based on the leaf blight severity level, the GWAS did not generate any SNP locus that passed the Bonferroni threshold. Six SNPs were found to be statistically significant. The LD heatmaps in Figure 6 show SNP locus S09_38670567 and an LD around the locus, indicating a low LD.

## 3. Discussion

*E. turcicum* is an important pathogen of sorghum, causing sorghum leaf blight [8,17]. Figure 7 shows leaf blight-infected sorghum leaves characterized by long elliptical reddish-purple or yellowish-tan lesions. A survey of 206 sorghum farmers’ fields in seven regions across Senegal during the 2019 growing season revealed that leaf blight was the most prevalent disease. Similarly, a total of 122 sorghum farmers’ fields surveyed across the same seven regions in Senegal during the 2022 growing season recorded a 100% prevalence of leaf blight [22]. Among the accessions evaluated against the leaf blight pathogen, N15 from Niger exhibited the lowest incidence and severity, indicating that this accession may possess resistance genes to *E*. *turcicum*. One sorghum line, “Hageen Durra”, was found to be resistant to leaf blight when evaluated in Central Sudan [12]. In India, many accessions, such as IS 13870, IS 18758, and IS 19670, exhibited high to moderate levels of resistance to leaf blight when evaluated in the field [23]. In this study, top SNPs passing the Bonferroni threshold, identified from the GWAS and their associated genes for a leaf blight incidence rate, were noted. However, based on the leaf blight severity level, the GWAS did not generate any SNP locus that passed the Bonferroni threshold. In a recent study, a transcriptional analysis revealed that rhamnogalacturonate lyase, plantacyanin, and zinc finger were expressed over a 70-fold change when inoculated with *E. turcicum* in sorghum [10]. In a maize GWAS study, multiple genes associated with SANT, WRKY, inositol-pentakis-phosphate 2-kinase, potassium uptake protein TrkA, and 4′phosphopante theinyl transferase were listed as top candidates against *E. turcicum* [24]. Lipps et al. [25] used two sorghum recombinant-inbred-line (RIL) populations for resistance to *E. turcicum* and identified a total of six quantitative trait loci (QTLs) across the two populations. In this study, we examined *E. turcicum* resistance to over one hundred accessions originally collected from Niger and Senegal and evaluated in three locations in Senegal. The accessions tested in Ndiaganiao showed high incidence and severity levels, indicating that Ndiaganiao provided a favorable environment for *E. turcicum* infection. Among the accessions, the lowest incidence rates were observed in N15, N48, and S17, while several accessions, including N15, N43, N38, N46, N30, N28, and N23, exhibited the lowest severity levels. As expected, the correlation test proved a clear positive correlation between the two traits.

A GWAS identified six novel SNPs associated with an average incidence rate. Sobic.009G099450, a gene annotated as a zinc finger, was 22,641 bp away from the top SNP locus S09_38670567. As the SNP locus is in the LD with other SNPs in 2.2 Mb (SNPs in pairwise r^2^ > 0.3 with the significant SNP marker S09_38670567 span a 2.21 Mb region, while SNPs in pairwise r^2^ > 0.5 span 194 bp), it is unsurprising that the nearest annotated gene is 22,641 bp away from the SNP. Zing-finger-related SNPs were identified as top candidates in sorghum against fungal pathogens such as the Senegalese and sorghum mini-core collection [26,27,28]. CCHC-type zinc-finger proteins play key roles in both abiotic and biotic stress responses and plant growth and development [29]. S03_67084062 was closely located to the homeobox-leucine zipper (HD-ZIP) protein. Homeodomain leucine zipper proteins are plant-specific transcription factors that contain a homeodomain and a leucine zipper domain and are highly associated with abiotic stress [30]. Gene silencing (S08_9250700) is one of the critical defense mechanisms that protect plants from pathogens, and it controls the sequence-specific regulation of gene expression [31]. The closest annotated gene from S09_5849212 is associated with Acyl-CoA dehydrogenase 4, Peroxisomal. The biosynthesis of jasmonic acid in plant peroxisomes requires the action of acyl-coenzyme A oxidase [32]. The nearest gene of SNP locus S02_37426140 is the MYB/SANT-like DNA-binding domain. MYB transcription factor genes are well known to be involved in controlling various processes, like responses to biotic and abiotic stresses, development, differentiation, metabolism, and defense [33]. Except for SNP locus S05_48064154, all five SNPs associated with a leaf blight incidence rate were associated with plant defense and stress responses. 

Among the top candidate genes, Sobic.008G070700 and Sobic.009G057200 were available for the predicted protein network nodes (Figure 8). Sobic.008G070700 was associated with RNA polymerases, translation initiation factor 2c, and small RNA 2′-o-methyltransferase. Sobic.009G057200 was linked to 3-ketoacyl-CoA thiolase 2, enoyl-coa hydratase, acyl-coenzyme an oxidase 2, malonate-semialdehyde dehydrogenase, peroxisomal fatty acid, dihydrolipoamide acetyltransferase, and glyoxysomal fatty acid. Among the genes linked to our candidates, Arabidopsis 3-ketoacyl-CoA thiolase 2 is involved in abscisic acid (ABA) signal transduction, which plays a major role in plant defense [34]. In *Magnaporthe oryzae*, methylmalonate-semialdehyde dehydrogenase regulates conidiation, polarized germination, and pathogenesis [35]. Peroxisomal fatty acid β-oxidation is reported to negatively impact plant survival under salt stress [36]. Another study concluded that peroxisomal-CoA synthetase is involved in fatty acid β-oxidation and pathogenicity in rice blast fungi [37]. Likewise, many genes linked to our top candidate genes through network nodes are also associated with plant stress and defense responses, indicating the importance of the identified genes in this study. Overall, limited GWAS work has been conducted on sorghum leaf blight, compared to Northern corn leaf blight (NCLB) in Maize [24,25,37,38]. A GWAS on NCLB caused by *Setosphaeria turcica* (perfect stage of *E. turcicum*) identified 22 SNPs significantly associated with NCLB in three tropical maize germplasm mapping panels [38]. Zhang et al. [17] detected 113 unique candidate genes in sorghum lines evaluated against *E*. *turcicum*, and when these genes were compared to those found in NCLB in maize, they concluded that they play a role in resistance to both crops. Using a GWAS on NCLB in maize, 81 genes were revealed, and 12 of these genes were shown to play a role in plant defense [24]. Recently, Lipps et al. [25] detected six QTLs across two sorghum recombinant inbred line populations when evaluated for resistance against the leaf blight pathogen. All these GWAS on sorghum and maize against *E*. *turcicum* will continue to enhance our knowledge of the resistance mechanisms in these two economically important crops, thereby increasing our food security globally.

## 4. Materials and Methods

*Study area*: The research fields were established in Senegal, West Africa, during the 2022 growing season. Senegal is located between latitudes 12°30′ and 16°30′ N and longitudes 11°30′ and 17°30′ W, with the drier northern part lying in the Sahelian zone and the southern part with more rain in the Sudanian zone [39]. The three fields were in Kaymor (13°58′33.7″ N and 15°05′03.2″ W) in the Kaolack region, Kolda (12°51′14.7″ N and 14°48′15.4″ W) in the Kolda region, and Ndiaganiao (14°35′58.7″ N and 16°44′10.7″ W) in the Thies region. The annual rainfall is 752 mm in Kaolack, 1072 mm in Kolda, and 377 mm in Thies, while the minimum and maximum temperatures range from 25 to 35 °C, 24 to 35 °C, and 23 to 33 °C, respectively [15]. These sites were selected because of their history of a high incidence of leaf blight, ranging from 88% to 100% [15,22]. A total of 120 sorghum accessions with bright non-diseased seeds were randomly collected from farmers′ fields in Niger and Senegal, and with SC748-5 and BTx623 were planted in these three different environments. Seeds of each accession were planted in 1.8 m rows with 0.8 m row spacing in each field environment. The accessions were planted in a randomized complete block design, and each accession was replicated thrice. The fields were kept weed-free with occasional hand-hoeing. Plants were evaluated for leaf blight incidence at the soft-to-early-hard-dough stage of development. The leaf blight incidence was based on the formula below [14,15]:$$Incidence=\frac{Number\;of\;plants\;with\;the\;disease\;in\;a\;row}{Total\;number\;of\;plants\;in\;a\;row}\times 100$$

Disease severity scale: The severity scale was previously described by Prom et al. [14] and spanned from 0 to 11 with mid-points, where 1 = 5.5, 2 = 15.5, 3 = 25.5, 4 = 35.5, 5 = 45.5, 6 = 55.5, 7 = 65.5, 8 = 75.5, 9 = 85.5, 10 = 95.5, and 11 = 100, used to calculate the mean severity. 

*Statistical Analysis*: Student’s *t*-test for all possible comparisons was performed across the population with JMP Pro 15 (SAS Institute, Cary, NC, USA). In addition, the incidence rate and severity were compared across the three experimental locations using Student’s *t*-test in JMP Pro 15. A Pearson correlation coefficient was computed to test the correlation between the incidence rate and severity of the disease as well. 

### GWAS

DNA extraction of fresh sorghum leaves sampled from 120 genotypes from Niger and Senegal (60 from each country) was conducted using either a NucleoSpin Plant II kit (Macherey-Nagel, ref. 740770) or a modified 2% Cethyl Trimethyl Ammonium Bromide (CTAB) method [40,41]. The DNA was then purified using 7.5 M ammonium acetate and isopropanol. The OD ratios (260/230, 260/280) were checked using a SpectraMax^®^ QuickDrop™ Micro Volume spectrophotometer (Molecular Devices), and the samples were run on a 1% agarose gel stained with ethidium bromide for quality control. The Genomics and Bioinformatics Service performed the sequencing at Texas A&M (TxGen, 1500 Research Pkwy Suite 250, College Station, TX 77845, USA) using an Illumina NovaSeq 6000 (2.6X average coverage). All the samples were repurified before library preparation. The sequencing raw data were processed using the GATK (Genome Analysis Tool Kit) best practices for variant calling, implemented in the DRAGEN platform (Illumina, San Diego, CA, USA). The sequences were aligned against Sorghum bicolor v3.1.1, available at Phytozome (https://phytozome-next.jgi.doe.gov/info/Sbicolor_v3_1_1, accessed on 17 October 2023), as the reference genome for SNP calling. The calls were filtered using the following parameters: minimum coverage depth = 3 and minimum genotype quality = 9 on a Phred scale. The resulting variants were then filtered to only accept SNPs with a minimum minor allele frequency of 0.05 and a maximum rate of missing data of 0.5. Variant filtration was performed using bcf tools. The SNP data were finally inputted using Beagle (Beagle 8.7e1.jar). The output was split by chromosome in a VCF format and contained over 5 × 10^3^ SNPs. PLINK v1.9 [42] was used for the VCF file conversions and a random selection of 50,000 SNPs from the genotypic data. An analysis of the population structure was performed in R studio v1.4.1717 [43], using the FactoMineR v2.8 [44] and Factoextra v1.0.7 [45] packages for a principal component analysis (PCA) and k-means clustering with the average silhouette method. For the validation and assignment of the accessions to genetic groups, a dendrogram was generated using SNPRelate v1.28.0 and gdsfmt v1.30.0 [46] and visualized with ggtree v3.2.1 [44]. The linkage disequilibrium (LD) of a local variant around the statistically significant SNP was plotted using LDheatmap v1.0-6 [47]. For the GWAS, we utilized GEMMA v0.98.3 [48]. Association tests were performed using a univariate linear mixed model, accounting for the relatedness matrix as a covariate term and employing the Wald test for determining statistical significance. To generate phenotype and genotype distribution, we utilized the software package vcf2gwas v0.8.3 [49]. Top candidate SNPs that surpassed the Bonferroni test were tracked to identify their exact location using the reference sorghum genome sequence, version 3.1.1, accessed through the JGI Phytozome 13 website. All available predicted protein network nodes from the top candidate genes were searched through the STRING database version 11.5 website (https://string-db.org/, accessed on 17 October 2023).

## 5. Conclusions

In Senegal, sorghum ranks third after millet and maize among dryland cereal production, while in Niger, sorghum ranks second to pearl millet in cereal production [4,5,50]. In both countries, sorghum plays a critical role in the daily lives of hundreds of millions of inhabitants for human food, drinks, baked food, animal feed, commercial ventures, building materials, and other uses. An increase in sorghum production will be critical in ensuring food security, especially in semi-tropical and tropical countries, including these two countries, due to the expected increase in global population and the impact of climate change. A robust and integrated management of sorghum diseases such as leaf blight and pests are components that must be addressed to realize the goal of increasing production by 2050, and any information that contributes to our understanding of the host–pathogen interaction will be of value. This work is significant because it suggests that the N15, N48, and S17 lines identified may be excellent sources for use in breeding for leaf blight resistance in both countries. In addition, the candidate genes identified in this study are anticipated to offer additional options for enhancing plant resistance against fungal pathogens through plant breeding or gene editing. Finally, the utilization of GWAS on sorghum accessions is one way to identify loci (SNPs) associated with leaf blight resistance, incited by *E. turcicum*, a widespread and devastating disease in West Africa, which is essential for breeding strategies aimed at predicting germplasm with disease resistance and high-yielding germplasm to ensure future food security in the region.

## Figures and Tables

**Figure 1 plants-12-04010-f001:**
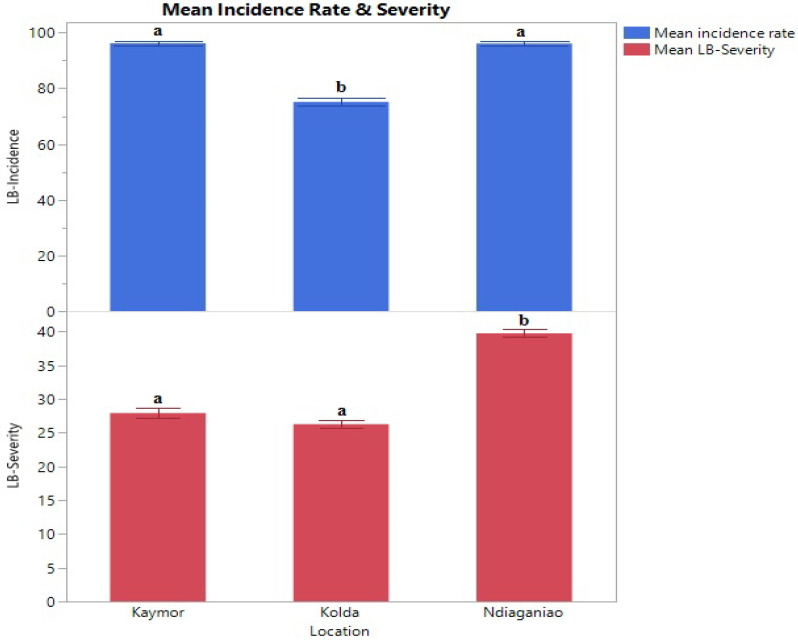
Comparisons of leaf blight incidence rate (blue) and severity level (red) based on three locations. Different letters indicate statistical significance.

**Figure 2 plants-12-04010-f002:**
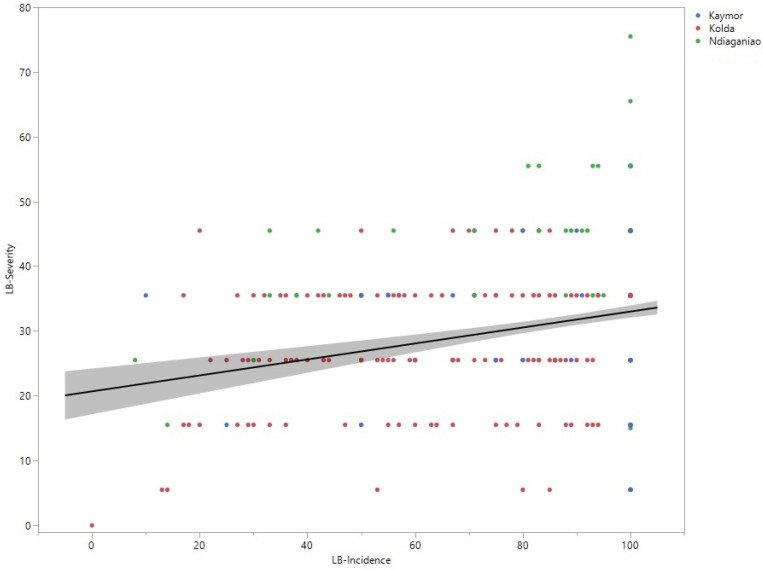
The correlation between leaf blight incidence rate and severity level. Pearson’s correlation showed a weak positive correlation of 0.22 with *p* < 0.0001. Different colors indicate surveyed locations in Senegal.

**Figure 3 plants-12-04010-f003:**
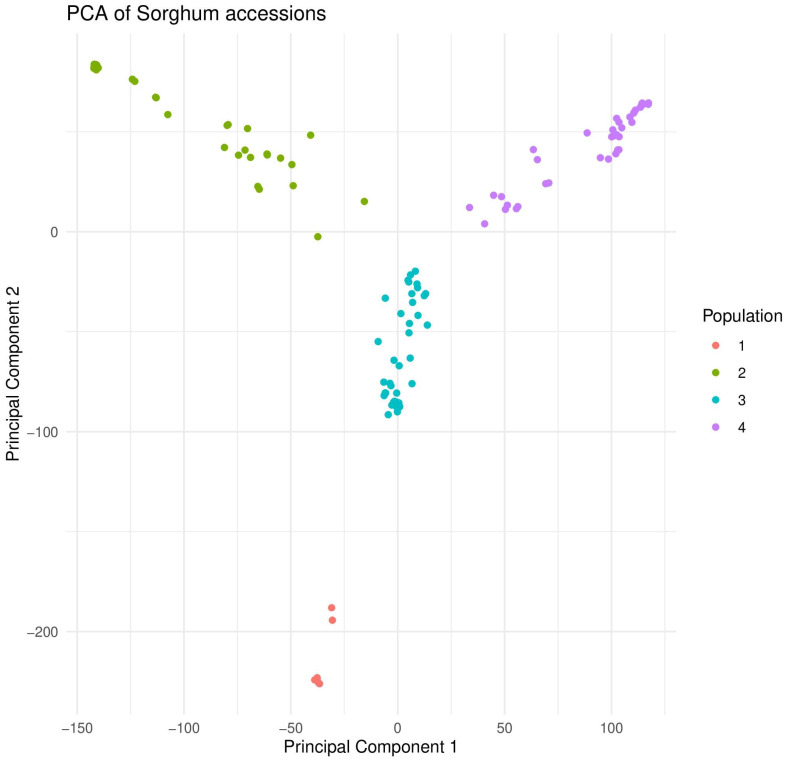
The principal component analysis of the Nigerien and Senegalese sorghum accessions. Four colors indicate different groups (red, green, cyan, and purple).

**Figure 4 plants-12-04010-f004:**
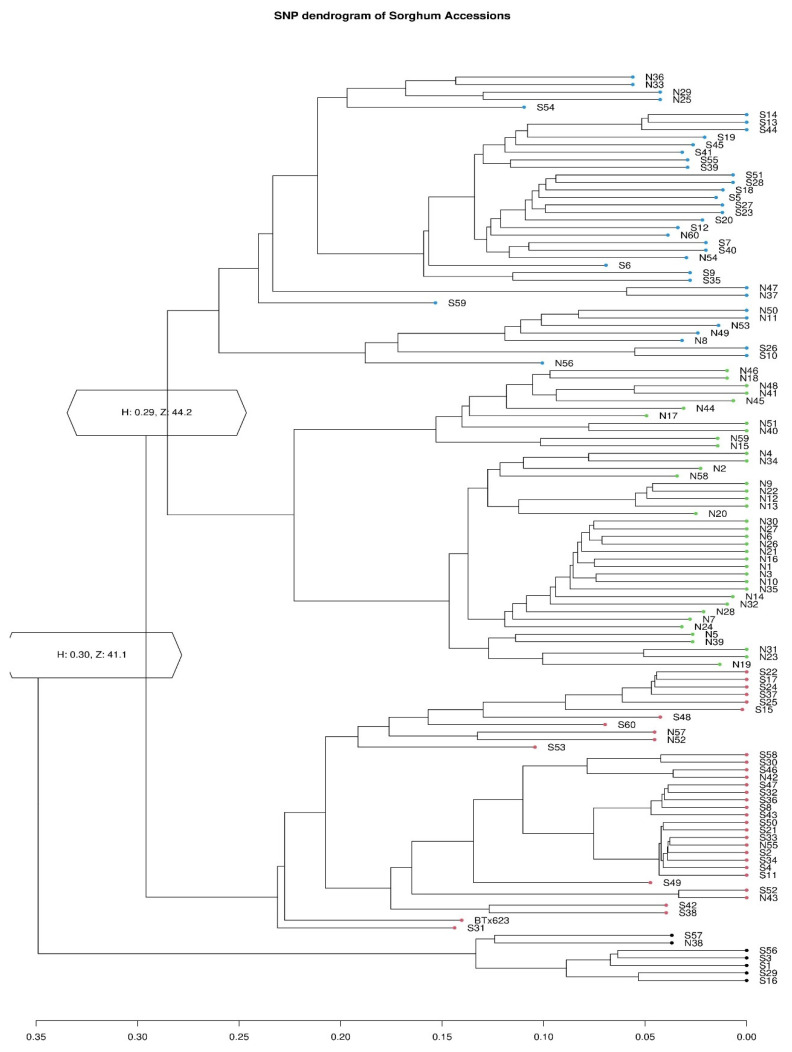
A dendrogram was constructed based on the genotype data of the accessions. Four major genetic groups were shown, indicated by the two main branches and different colors.

**Figure 5 plants-12-04010-f005:**
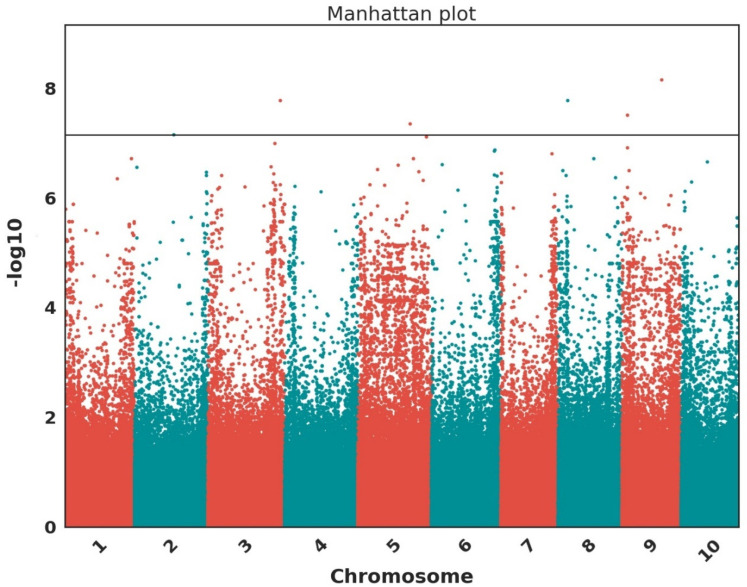
The genome-wide association for leaf blight incidence rate. Manhattan plot across ten chromosomes is shown with the Bonferroni threshold, indicating that six SNPs are statistically significant.

**Figure 6 plants-12-04010-f006:**
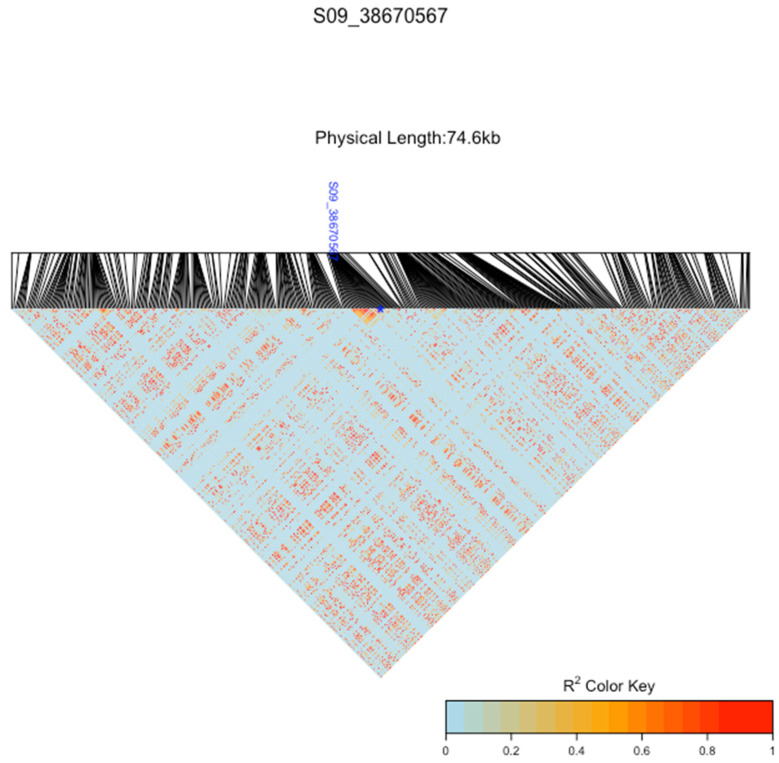
LD heatmaps visualize the LD of SNP locus S09_38670567. The blue stars indicate the exact location of the SNP locus.

**Figure 7 plants-12-04010-f007:**
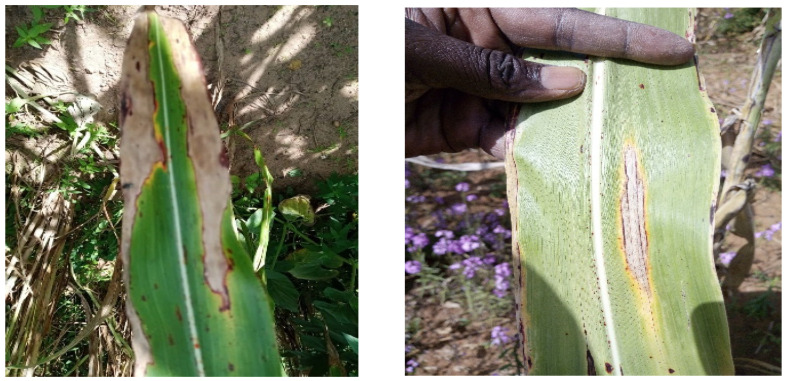
Sorghum leaves infected with leaf blight.

**Figure 8 plants-12-04010-f008:**
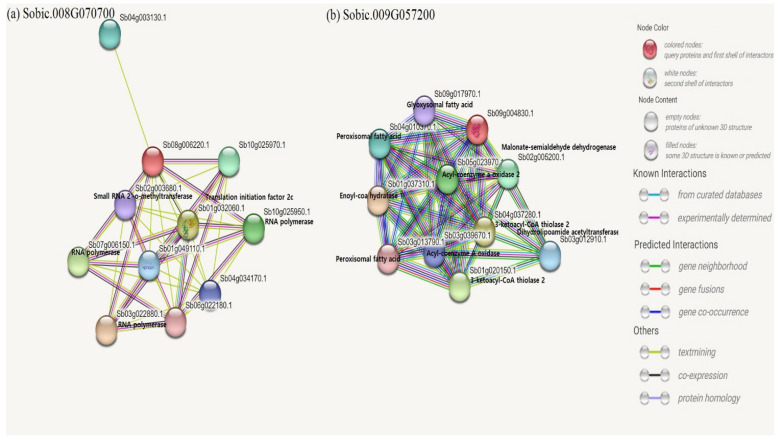
Available predicted protein network nodes of top candidate genes. (**a**) Sobic.008G070700. (**b**) Sobic.009G057200. Red beads indicate the candidate genes. Genes lacking annotated functions were shown, but no function was listed right next to each gene.

**Table 1 plants-12-04010-t001:** The average leaf blight incidence rate across the three locations ordered from high to low. The standard error of the mean is listed next to the average value. Single dots indicate cultivars tested only once.

Accessions	Incidence	SEM	Accessions	Incidence	SEM
S47	100	0	S31	88.14	7.88
S42	100	0	N49	88.14	7.88
N5	100	0	N4	88.13	6.31
N7	100	.	N19	88.13	10.65
N9	100	0	S8	88	6.77
S36	100	0	S3	88	9.07
N27	100	0	S1	88	4.68
N36	100	0	N46	87.89	5.42
S19	100	0	N57	87.14	8.48
N43	100	0	N26	87.13	8.58
S52	100	0	S12	86.86	8.53
N30	99.13	0.88	N50	86.78	7.70
S46	98.33	1.67	S55	86.5	7.53
S2	98.29	1.71	S29	86.44	6.19
S27	98	1.36	S14	86.11	8.59
N53	97.89	2.11	S38	85.75	7.84
S4	96.75	2.33	S51	85.43	7.18
S21	96.67	3.33	S37	85.25	8.51
S32	96.63	2.31	N18	85.13	7.67
S40	96	2.77	S23	85.13	7.52
N52	95.86	4.14	N28	84.29	10.48
SC748-5	95.86	4.14	N55	84.29	10.33
S11	94.71	3.43	S39	83.67	8.92
S30	94.5	5.5	N8	83.63	10.85
S58	94.5	5.5	N29	83.57	8.55
N25	94.44	3.91	S20	82.75	9.61
S59	94.13	4.24	S50	82	10.21
S49	94	3	N44	81.88	8.68
S60	94	3.98	S33	81.29	12.13
S57	93.89	4.55	S5	80.86	11.69
N23	93.44	5.52	N39	80.71	12.59
S34	93.25	4.44	N58	80.13	9.66
N54	93.22	5.02	N42	80	10.38
N20	92.22	5.21	N41	80	11.55
S44	92	3.98	S43	79.86	11.2
S18	92	5.89	S6	79.38	9.55
N34	91.44	4.76	N51	79.22	9.88
BTx623	91.14	7.06	N40	79	10.83
S22	91.11	8.89	N6	78.75	9.42
S56	90.89	5.33	S41	78.5	16.4
S48	90.86	6.22	N2	78.5	12.84
S10	90.38	5	N45	78.13	11.83
N24	90.33	7.1	S15	77.86	7.95
S35	90.25	5.6	S45	77.78	12.38
S28	89.38	4.2	N59	76.5	9.72
S16	89.13	10.88	N56	76.13	10.89
S7	89.13	6.63	S13	75.88	12.41
N60	89	7.2	N38	72.86	12.58
S9	88.89	7.35	S17	70.83	11.36
N22	88.71	9.44	N48	67.5	12.62
N3	88.67	5.8	N15	50	.
S54	88.66	7.09	Average	88.4	0.76

**Table 2 plants-12-04010-t002:** The average leaf blight severity level across the three locations ordered from high to low. The standard error of the mean is listed next to the average value. Single dots indicate cultivars tested only once.

Accessions	Severity	SEM	Accessions	Severity	SEM
S32	41.75	7.54	S38	31.75	4.98
S11	41.21	2.02	S47	31.75	3.24
S1	39.25	3.24	S59	31.75	3.24
S31	38.36	4.21	N2	31.25	6.96
S17	37.17	1.67	S2	31.21	2.97
S21	37.17	5.43	S33	31.21	6.12
N58	36.75	5.15	SC748-5	31.21	4.81
S18	36.75	3.98	N3	31.06	4.12
S34	36.75	3.98	N20	31.06	4.12
N40	35.51	3.33	N25	31.06	4.44
N7	35.5	.	N27	31.06	5.56
N42	35.5	4.36	N34	31.06	4.12
S4	35.5	3.78	N51	31.06	4.12
S15	35.5	3.78	S39	31.06	4.44
S35	35.5	2.67	N8	30.5	5.35
S46	35.5	6.32	N36	30.5	5.63
S58	35.5	2.58	N56	30.5	3.78
S56	34.39	2.61	S6	30.5	3.78
N19	34.26	5.15	S8	30.5	4.63
N18	34.25	2.95	S30	30.5	5.63
S23	34.25	2.95	S37	30.5	5
S28	34.25	2.95	S22	29.94	4.44
S5	34.07	4.59	S60	29.94	3.77
S48	34.07	5.95	N29	29.79	2.02
S51	34.07	4.59	N55	29.79	2.97
S41	33.83	4.77	S44	29.79	3.69
S42	33.83	6.54	S52	29.79	3.69
S9	33.28	3.64	N5	29.25	5.96
S40	33.28	3.24	N41	29.25	4.6
S57	33.28	3.64	N9	28.83	8.03
N59	33	3.66	N39	28.36	5.22
S7	33	6.75	N49	28.36	5.65
S20	33	3.66	N52	28.36	5.22
S55	33	4.12	S12	28.36	5.22
N22	32.64	3.6	N6	28	5.59
N38	32.64	2.86	N26	28	5.59
N57	32.64	3.6	N44	28	3.66
S36	32.64	5.22	N24	27.72	4.65
S43	32.64	2.86	S14	27.72	3.64
N50	32.17	5.27	BTx623	26.86	3.44
N53	32.17	4.41	S16	26.75	2.27
N54	32.17	3.73	S49	26.75	5.15
S19	32.17	4.08	S29	26.61	3.89
S27	32.17	3.33	S45	26	5.46
S50	32.17	4.08	N23	25.5	5.53
S54	32.17	4.08	N28	25.5	4.88
N4	31.75	4.6	N30	25.5	5
N45	31.75	4.6	N46	25.5	4.41
N60	31.75	4.2	N48	24.25	5.15
S3	31.75	3.75	N43	21.21	3.69
S10	31.75	4.6	N15	15.5	.
S13	31.75	4.2	Average	31.57	0.43

**Table 3 plants-12-04010-t003:** Annotated genes nearest to the most significant SNPs are associated with leaf blight incidence rate. All six SNPs passed the Bonferroni threshold.

Chr	Location	Candidate Gene and Function	Base Pairs	Allele	*p*-Value
9	38670567	Sobic.009G099450 Zinc finger, CCHC-type	22,641	Reference: C Alternate: G	0.000000007
3	67084062	Sobic.003G351501 Homeobox-leucine zipper protein anthocyaninless2-related	1083	Reference: G Alternate: A	0.000000017
8	9250700	Sobic.008G070700 Protein suppressor of gene silencing 3 XS domain-containing protein/XS zinc finger domain-containing protein-related	15,203	Reference: A Alternate: T	0.000000017
9	5849212	Sobic.009G057200 Electron transport oxidoreductase//Acyl-CoA dehydrogenase 4, Peroxisomal	4543	Reference: C Alternate: T	0.000000031
5	48064154	No annotated gene nearby	-	Reference: T Alternate: A	0.000000044
2	37426140	Sobic.002G147900 MYB/SANT-like DNA-binding domain	52,147	Reference: G Alternate: A	0.00000007

## Data Availability

The data presented in this study are available upon reasonable request from the authors L.K. and C.W.M.

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
