# Peer review of "A Genome-Wide Association Study of Nigerien and Senegalese Sorghum Germplasm of Exserohilum turcicum, the Causal Agent of Leaf Blight"

_plants, 2023, doi:10.3390/plants12234010_

Round 1

Reviewer 1 Report

Comments and Suggestions for Authors

In their submitted paper, Prom and colleagues conducted an analysis of sorghum resistance to Exserohilum turcicum. In general, I find the study's design to be appropriate, but I do have a few comments:

  1. The description of the results is rather concise. I believe that the paper would benefit from an expanded discussion of the findings.

  2. It might be advisable to relocate Table 1 to the supplementary section.

  3. It would be more conventional to initiate the results section with a description of the population structure, including PCA and dendrogram analysis, before delving into the specifics of the resistance to Exserohilum turcicum.

  4. Consider replacing the bar plot with a box or violin plot to better illustrate the distribution of both incidence and severity.

  5. Given the non-normal distribution of incidence and severity, it may be preferable to use Spearman correlation analysis in such cases.

  6. The authors mention that each accession was replicated three times in each location. Does this imply the presence of three blocks in each location? If so, it would be possible to present a broad-sense heritability analysis. Otherwise, at the very least, narrow-sense heritability could be presented and discussed.

Author Response

Reviewer 1

  1. The description of the results is rather concise. I believe that the paper would benefit from an expanded discussion of the findings.

Authors’ response: The discussion was expanded as highlighted in Section 3.

  1. It might be advisable to relocate Table 1 to the supplementary section.

Authors’ response: Since the table is one of the most important contents of the manuscript, we decided to put it in the main manuscript as it is.

  1. It would be more conventional to initiate the results section with a description of the population structure, including PCA and dendrogram analysis, before delving into the specifics of the resistance to Exserohilum turcicum.

Authors’ response: We understand the reviewer’s point, but as the purpose of the study is identifying associations between the disease responses to genetic markers, we believe the way it is presented is more suitable for the purpose.

  1. Consider replacing the bar plot with a box or violin plot to better illustrate the distribution of both incidence and severity.

Authors’ response: We understand the reviewer’s point, but we believe the bar graph is simpler and easy to be understood by readers.  

  1. Given the non-normal distribution of incidence and severity, it may be preferable to use Spearman correlation analysis in such cases.

We tested both Spearman’s and Pearsons, but the differences between the two tests were neglectable.

  1. The authors mention that each accession was replicated three times in each location. Does this imply the presence of three blocks in each location? If so, it would be possible to present a broad-sense heritability analysis. Otherwise, at the very least, narrow-sense heritability could be presented and discussed.

Authors response: Yes, we had three blocks in each location; however, the aim of the study was to identify chromosomal locations associated with leaf blight response. Nevertheless, in a subsequent paper, the broad-sense heritability analysis could be explored.

Reviewer 2 Report

Comments and Suggestions for Authors

This study provides Genome-wide association studies of Nigerien and Senegalese sorghum germplasms to Exserohilum turcicum, the causal agent of leaf blight. The findings showed disease severity at different environments and position of some candidate genes on chromosomes. However, there are some limitations which must be addressed.

The abstract should provide some specific results such as quantitative results.

Also provide where maximum severity was observed.

Conclusion statement is missing in the abstract.

Line 43-44 which studies shown?

In introduction provide the host range and specific environment required for the pathogen.

Also relate it with study area.

How GWAS can be helpful to tackle the problem.

Provide background and literature review of the GWAS sorghum germplasms to Exserohilum turcicum.

Line 63-64 lack references and must be cited with recent studies such as doi: 10.1021/acs.jafc.3c02415, doi: 10.1038/s41467-022-32364-3, https://doi.org/10.1016/j.plaphy.2021.01.042

Line 64-67 should me more refine and clear as it is objective of the study. Also provide hypothesis or novelty of the study in last paragraph.   

“ Four colors indicate different groups” write names of the groups.

The authors should present environmental conditions of the studied regions or sampling sites.

On which basis only these three regions were selected for sampling? What are the reasons?

Exserohilum. Turcicum” remove . from the names.

It would be better to provide some images of the diseased samples to show disease symptoms to readers.

Why the author used word Niger in some places and Nigeria in some places? It must be consistent.

Line 211-213 the collected samples were fresh or diseased? Mention it

 Section 4 how disease was applied to the accessions provide complete protocol, which kind of accessions were preferred for collection?

“planted in these three different environments” specify the environments or provide the details.

Where the experiment was conducted green house or open field? Also provide conditions and protocol of the plantation.

In the samples, are there any resistant varieties to Exserohilum Turcicum?

Comments on the Quality of English Language

Some sentences are very long and not clear as mentioned in the comments

Author Response

Reviewer 2

The abstract should provide some specific results such as quantitative results. 

Authors’ response. Agreed. We included the percentage of the lines with lowest incidences in the abstract section. Lines

Also provide where maximum severity was observed.

Authors response: Noted in the results section (First paragraph).  The highest severity was observed in Kaymor.

Conclusion statement is missing in the abstract. 

Authors response: Agreed. Conclusion statement included in the abstract.

Line 43-44 which studies shown? 

Authors response: These studies are cited [8,9,10].

In introduction provide the host range and specific environment required for the pathogen.

Authors response: Host range and weather conditions that favor disease development are highlighted in the introduction. Lines

Also relate it with study area; The authors should present environmental conditions of the studied regions or sampling sites. On which basis only these three regions were selected for sampling? What are the reasons?

Authors’ response to comments and/or suggestions to 7, 13, and 14 is noted in the Materials and Methods section highlighted in lines……………..

How GWAS can be helpful to tackle the problem?. 

Authors’ response: The significant of this study is noted in the last paragraph of the conclusion section. Highlighted in lines….

Provide background and literature review of the GWAS sorghum germplasms to Exserohilum turcicum.

Authors response:  The last paragraph in the Discussion section lines……highlighted some of the previous GWAS conducted on maize and sorghum. Lines…..

Line 63-64 lack references and must be cited with recent studies such as doi: 10.1021/acs.jafc.3c02415, doi: 10.1038/s41467-022-32364-3, https://doi.org/10.1016/j.plaphy.2021.01.042

Authors’ response:  Sentence cited with recent articles published.

Line 64-67 should me more refine and clear as it is objective of the study. Also provide hypothesis or novelty of the study in last paragraph.

Authors’ response: A hypothesis was noted and the objective was amended.

Four colors indicate different groups” write names of the groups.

Authors’ response: Modification was made as suggested.

Exserohilum. Turcicum” remove  from the names.

Authors’ response:  Where do we need to remove this name?

It would be better to provide some images of the diseased samples to show disease symptoms  to readers. 

Authors’ response: Figure 7. Show leaf blight infected sorghum.

Why the author used word Niger in some places and Nigeria in some places? It must be consistent.

Authors’ response:  Agreed.  Nigeria deleted.  Text amended as follow: Niger or Nigerien used throughout the text.

Line 211-213 the collected samples were fresh or diseased? Mention it.

Authors’ response:  Yes. It is mentioned in the first line. DNA extraction of fresh sorghum leaves sampled from 120 genotypes……..(CTAB) method.

Section 4 how disease was applied to the accessions provide complete protocol, which kind of accessions were preferred for collection? 

Authors’ response: Noted in Section 4: sorghum accessions with bright non-diseased seeds were arbitarily collected from famers’ fields in Niger and Senegal.

“planted in these three different environments” specify the environments or provide the details.

Authors’ response:  Environmental conditions of the experimental sites noted in Section 4.

Where the experiment was conducted green house or open field? Also provide conditions and protocol of the plantation. 

Authors’ response:  Noted in Section 4.

In the samples, are there any resistant varieties to Exserohilum Turcicum?

Authors’ response:  Accession N15 from Niger exhibiting the lowest incidence and severity, indicating that it may possess resistance genes to E. turcicum.

Round 2

Reviewer 2 Report

Comments and Suggestions for Authors

The authors revised the study but still some points are missing and should be considered to improve the study.

The study does not mention the number of replications or the experimental design used in evaluating the sorghum accessions. Including multiple replications would provide statistical robustness and allow for a better assessment of the consistency and reliability of the results.

The study mentions that the accessions were sequenced and a genome-wide association study (GWAS) was conducted. However, it does not provide detailed information on the sequencing methods, quality control measures, or the statistical analysis procedures used in the GWAS. Providing more information on the genetic analysis would enhance the transparency and reproducibility of the study.

Line 71-73 additional information is required such as how genome wide studies would be helpful for the improvement of the germplasms.

And specify what are recent advances specifically to biotic stresses.

Line 70-72 should be cited with some more studies such as the following studies https://doi.org/10.3389/fgene.2021.635043, doi: 10.1038/s41467-022-32364-3

GWAS if one time is abbreviated should not be used full form after that. Like genome-wide association studies (GWAS) of the incidence and severity of Nigerien and Senegalese….. GWAS play important role… the study concluded that GWAS… revise in the whole MS. These are just examples. See other abbreviations as well.

Comments on the Quality of English Language

GWAS if one time is abbreviated should not be used full form after that. Like genome-wide association studies (GWAS) of the incidence and severity of Nigerien and Senegalese….. GWAS play important role… the study concluded that GWAS… revise in the whole MS. These are just examples. See other abbreviations as well.

Author Response

We are grateful to you and the reviewer for your comments and suggestions.  The manuscript was amended as required.

Reviewer: The study does not mention the number of replications or the experimental design used in evaluating the sorghum accessions. Including multiple replications would provide statistical robustness and allow for a better assessment of the consistency and reliability of the results.

Our answer: Thank you so much for the suggestion. Lines 234- 239 answered your comments on the experimental design and the number of replications at each location.  This information was in the original and revised version of the manuscript.

Reviewer: The study mentions that the accessions were sequenced and a genome-wide association study (GWAS) was conducted. However, it does not provide detailed information on the sequencing methods, quality control measures, or the statistical analysis procedures used in the GWAS. Providing more information on the genetic analysis would enhance the transparency and reproducibility of the study.

Our answer: Lines 249- 290 answered your comments on sequencing, statistical analysis, and GWAS portions. This information was in the original and revised version of the manuscript. A more detailed description was added.

Reviewer:

  1. Line 71-73 additional information is required such as how genome wide studies would be helpful for the improvement of the germplasms. And specify what are recent advances specifically to biotic stresses.
  2. Line 70-72 should be cited with some more studies such as the following studies https://doi.org/10.3389/fgene.2021.635043, doi: 10.1038/s41467-022-32364-3

Our answer: Thank you for the suggestions. We have added other GWAS studies in the text (Tomar 2021; Pradhan et al. 2020;Liu et al 2017; Kawicha et al. 2023) as suggested.

Reviewer:

GWAS if one time is abbreviated should not be used full form after that. Like genome-wide association studies (GWAS) of the incidence and severity of Nigerien and Senegalese….. GWAS play important role… the study concluded that GWAS… revise in the whole MS. These are just examples. See other abbreviations as well.

Comments on the Quality of English Language

GWAS if one time is abbreviated should not be used full form after that. Like genome-wide association studies (GWAS) of the incidence and severity of Nigerien and Senegalese….. GWAS play important role… the study concluded that GWAS… revise in the whole MS. These are just examples. See other abbreviations as well.

Our answer: Thank you for the suggestions. The two suggestions are essentially identical. The full name for GWAS was introduced in the abstract, and after that, it was mentioned in the introduction part: “Thus, genome-wide association studies (GWAS) of the incidence and severity of Nigerien and Senegalese sorghum germplasms planted in three environments in Senegal evaluated against the leaf blight pathogen were conducted.” As GWAS was first introduced in the main manuscript, we think it should be a full name. After that, GWAS was never mentioned in full name in the main manuscript. Other abbreviated words were carefully used, and modifications were made as suggested.

Round 3

Reviewer 2 Report

Comments and Suggestions for Authors

NA